# A Facile Strategy for Development of pH-Sensing Indicator Films Based on Red Cabbage Puree and Polyvinyl Alcohol for Monitoring Fish Freshness

**DOI:** 10.3390/foods11213371

**Published:** 2022-10-26

**Authors:** Hejun Wu, Chun Jiao, Shasha Li, Qingye Li, Zhiqing Zhang, Man Zhou, Xiangyang Yuan

**Affiliations:** 1College of Science, Sichuan Agricultural University, Ya’an 625014, China; 2College of Food Science, Sichuan Agricultural University, Ya’an 625014, China

**Keywords:** red cabbage puree, anthocyanins, polyvinyl alcohol, pH sensitivity, intelligent packaging, fish freshness

## Abstract

This study aimed to develop a novel pH-sensing biopolymer film based on red cabbage puree (RCP) incorporated with polyvinyl alcohol (PVA), which was utilized for monitoring fish freshness during storage at 25 °C. A homogenized RCP suspension with a mean particle size of 12.86 ± 0.03 μm and a total anthocyanin concentration of 292.17 ± 2.65 mg/L was directly used as a film-forming substance and anthocyanin source to blend with PVA, showing visual changes in color and ultraviolet-visible spectra within a pH of 2–12. Rheological and microstructural studies certified the strong interactions and good compatibility between the RCP and PVA, resulting in better mechanical properties and water resistance of the composite film than those of a pure RCP film, but without affecting its pH sensitivity. When used for fish freshness monitoring at 25 °C, the developed RCP/PVA film presented visible color differences from purple to yellow, which corresponded to the spoilage threshold of the total volatile basic nitrogen and the total viable count in fish samples. The study highlights that anthocyanin-rich purees of fruits and vegetables, in this case red cabbage puree, can be fully utilized to develop eco-friendly pH-sensing indicator films for intelligent food packaging.

## 1. Introduction

Nowadays, technological innovations in food-packaging systems, including active packaging, intelligent packaging, and biodegradable/edible packaging, have been developed so as to maintain and monitor food safety and quality, prolong shelf-life, and reduce the environmental burden of plastic waste [1]. Among these, intelligent packaging is regarded as one emerging concept with huge potential in the food industry, since it can integrate intelligent functions with conventional packaging systems by using sensors or indicators, which are capable of monitoring, sensing, recording, tracing, and conveying information reflecting the safety, quality, and integrity of food [2,3]. Among various indicators for intelligent food packaging systems, pH indicators have been developed to monitor the freshness of food, since pH changes are indicative of spoilage in many food products, and they are highlighted due to their sensitivity, efficiency, simplicity, and low cost [4,5].

In general, pH indicators are composed of a pH-sensitive colorant and a solid matrix to immobilize it, exhibiting visible color changes under acidic and alkaline conditions for intelligent packaging applications [6]. Currently, increasing attention has been paid to the development of pH-sensitive indicator films derived from natural dyes and biopolymers because of their non-toxicity, abundance, and biodegradability [7,8,9,10]. In particular, anthocyanins are natural water-soluble pigments with very attractive colors found in several red/purplish fruits, vegetables, and flowers that also possess many biological activities with potential health benefits in addition to their color attributes [10,11,12]. Among the anthocyanin sources, red cabbage stands out not only because of its high substrate concentration and intense coloration, but also for its unique anthocyanins with color changes over a very broad pH range [11,12]. The anthocyanins extracted from red cabbage can change color from red at a low pH to blue and green at a high pH, making them an attractive alternative to synthetic chemical dyes for the fabrication of natural pH-sensitive indicators [13]. For example, Prietto et al. [14] developed pH-sensitive films composed of corn starch and anthocyanins extracted from either black bean seed coats or red cabbage, and found that the pH-sensitive films with red cabbage anthocyanins presented greater potential for the development of intelligent packaging due to their higher stability during storage and greater color variation among pH levels. So far, red cabbage anthocyanins have been incorporated into different natural and/or synthetic biopolymers to develop intelligent films, including nanocellulose [13], starch [14], gelatin [15,16], konjac glucomannan [17], chitosan/starch [18], *Artemisia sphaerocephala* Krasch. gum/carboxymethyl cellulose sodium [19], chitosan/polyvinyl alcohol (PVA) [20], starch/PVA [21], and PVA/sodium carboxymethyl cellulose [22]. It is important to note that the properties of biodegradable/edible films derived from natural resources still need improvements, especially to their water-resistance and mechanical strength [23]. In view of this, many researchers have used PVA, a non-toxic and water-soluble synthetic polymer with excellent biodegradable, chemical resistance, film-forming, and mechanical properties, in the preparation of blends and composites with other renewable biopolymers for intelligent films containing anthocyanins with the appropriate properties [20,21,22,23,24,25,26,27,28].

However, most existing studies have mainly focused on the development of such intelligent films by firstly extracting anthocyanins from diverse sources and subsequently immobilizing them in various biopolymer matrices. The extraction procedures can be quite time consuming and can require expensive or even toxic organic solvents, including ethanol, methanol, and acetone [11]. Furthermore, residues are generated during extraction, which might be contrary to the increasing trend towards the full utilization of the whole plant [29]. Therefore, the exploration of intelligent pH-sensitive films containing anthocyanins with a low cost and a high efficiency is still worth studying. Instead of using extracts from fruits and vegetables, their juices or even their processing wastes or residues have been recently used for the production of intelligent films with a pH sensitivity [30,31,32]. For instance, Luchese et al. [32] directly used blueberry residue powder as an anthocyanin source to obtain starch-based pH indicator films that could monitor the spoilage of chicken meat. Nevertheless, the authors of this study concluded that these films could be commercially unaccepted and limited in their practical use due to their heterogeneous visual appearance and low tensile strength (<1.5 MPa), and thus, more investigation was required in order to verify their real applicability.

Because film-forming biomacromolecules (mainly including cellulosic and pectic substances) and other components (i.e., natural pigments and phenolic compounds) exist naturally in many fruits and vegetables, there are numerous reports of biodegradable/edible films made from their purees or residues, which have been well reviewed recently [29,33]. However, as pointed out by Otoni et al. [29], films comprised exclusively of fruits and vegetables may also frequently exhibit poor consistency, mechanical strength, and barrier properties; therefore, binding agents are required to improve the films’ physical properties. Likewise, PVA was found to enhance the mechanical properties, water resistance, and thermal stability of biodegradable films based on sugar beet pulp due to the good interaction and compatibility between them [34]. It has been reported that cabbages contain large amount of carbohydrates, including dietary fiber and many bioactive compounds [35]. More importantly, red cabbage is also an edible source of natural pigments with a high anthocyanin content (≥10 g/kg DM), and more than 30 anthocyanins have been detected in red cabbage extracts [36]. These features could make red cabbage a promising raw material for the development of biodegradable or edible films with additional functionality. 

Therefore, based on these considerations, the main objective of this study was to develop a novel, intelligent, pH-sensing biopolymer film by blending red cabbage puree (RCP) with PVA, which has not been reported yet as far as we know. Notably, RCP has been considered as both a film-forming substance and an anthocyanin source for pH-sensitive indicator films without prior extraction to realize its full high-value-added utilization. The effects of the incorporation of PVA as a binding agent on the rheological properties of film-forming suspensions (FFSs), as well as the microstructure, physicochemical properties, and pH sensitivity of the composite film, were assessed. Meanwhile, the potential application of the developed intelligent film for monitoring the freshness of fish during storage at 25 °C for 48 h was evaluated.

## 2. Materials and Methods

### 2.1. Materials

Red cabbage was purchased from a local supermarket (Ya’an, China). Polyvinyl alcohol (PVA 1799, with an average degree of polymerization: 1700 ± 50 and degree of hydrolysis: 99%) was supplied by Kelong Chemical Reagent Co., Ltd. (Chengdu, China). Plate count agar (PCA) was purchased from Hangzhou Microbial Reagent Co., Ltd. (Hanzhou, China). Other chemicals of an analytical grade, including glycerol, sodium hydroxide (NaOH), hydrochloric acid (HCl), methyl red, methylene blue, magnesium oxide, boric acid, and sodium chloride, were all procured from Kelong Chemical Reagent Co., Ltd. (Chengdu, China). 

### 2.2. Preparation of FFSs

A total of 200 g of red cabbage was added into 300 mL of distilled water, which was firstly squeezed by a high-speed juice blender (JYL-G12E, Joyoung Company Limited Co., Ltd., Jinan, China) at 15,000 rpm for 5 min, and then further homogenized by a high-pressure homogenizer (GJJ-0.03/100, Shanghai Noni Light Industry Machinery Co., Ltd., Shanghai, China) at a controlled pressure of 60 MPa for 10 passes. The homogenized RCP dispersion was obtained for the characterization and subsequent preparation of the films. A PVA solution was obtained by dissolving 10 g of PVA in 90 mL of distilled water under magnetic stirring at 90 °C for 1 h. An RCP/PVA suspension (RCP:PVA = 95:5, g/g) was then acquired by blending 95 g of the RCP dispersion with 5 g of the PVA solution, which was stirred constantly at 500 rpm for 30 min at room temperature to achieve complete mixing.

### 2.3. Characterization of FFSs

#### 2.3.1. Particle Size Distribution (PSD)

The particle size distribution of the RCP suspension was analyzed by a laser diffraction particle size analyzer (Rise-2006, Runzhi Technology Co., Ltd., Jinan, China) across a dynamic range of 0.1–2000 μm. Prior to the analysis, the sample was well shaken to ensure the sample’s homogeneity. Subsequently, the sample was diluted and dispersed in deionized water, and a refractive index of 1.33 for water was selected.

#### 2.3.2. pH

The pH of the RCP suspension was measured using a digital pH meter (PHS-25, Chengdu Century Ark Technology Co., Ltd., Chengdu, China).

#### 2.3.3. Total Anthocyanin Concentration (TAC)

The total anthocyanin concentration of the RCP suspension was determined by using the pH differential method according to Fuleki and Francis [37]. The detailed procedure for this measurement has been described by Musso et al. [16] in previous work. Briefly, the RCP suspension was centrifuged at 4000 r/min for 5 min to remove impurities, and 200 μL of the supernatant was added separately to 7 mL of a potassium chloride solution (0.025 M, pH = 1) and a sodium acetate solution (0.4 M, pH = 4.5). Then, their absorbance was measured at 530 and 700 nm using a multi-mode microplate reader (Varioskan Flash, Thermo Fisher Scientific, Waltham, MA, USA). Considering that red cabbage anthocyanins are all derived from cyanidin glycoside, the TAC was determined as mg of cyanidin-3-glucoside/100 mL of RCP suspension using Equations (1) and (2):(1)$$\Delta A={\left(A_{530}-A_{700}\right)}_{pH=1}-{\left(A_{530}-A_{700}\right)}_{pH=4.5}$$
(2)$$\;\mathrm{TAC}=\frac{\Delta A\times MW\times DF\times 1000}{\varepsilon \times L}\;\;\;\;\;$$
where ∆*A* is the difference in absorbance change, *MW* is the molecular weight of cyanidin-3-glucoside (449.2 g/mol), *DF* is the dilution factor, *ε* is the molar extinction coefficient of cyanidin-3-glucoside (26,900 L/cm mol), and *L* is the path length (cm).

#### 2.3.4. UV−Vis Spectra Measurement

The pH of the RCP and RCP/PVA suspensions was adjusted by 1 M HCl or 1 M NaOH to a range of 2–12. The UV–Vis spectra of the RCP and RCP/PVA suspensions at different pH values were measured using a multi-mode microplate reader (Varioskan Flash, Thermo Fisher Scientific, Waltham, MA, USA) in the range of 400–800 nm. The images of the samples were captured to correspond to their UV–Vis spectra.

#### 2.3.5. Rheological Properties

The rheological properties of the PVA solution, RCP, and RCP/PVA suspension were characterized by a rheometer (DHR-1TA, TA Instruments, New Castle, DE, USA) equipped with a parallel-plate geometry (40 mm diameter, 0° angle, and 1.0 mm gap). The shear rate varied from 0.1 to 100 s^−1^ at 25 °C. The storage or elastic modulus (G′) and loss or viscous modulus (G″) were measured from 0.1 to 100 rad/s using an angular frequency sweep at a strain amplitude of 1% to analyze the dynamic rheology.

### 2.4. Preparation of the Films

According to the preliminary experiments, FFSs (obtained as described in Section 2.2) were mixed with 0.3% glycerol (*w*/*w*, based on the mass of the FFS) as a plasticizer by stirring for another 30 min. After vacuum degassing at −0.1 MPa for 1 h, 100 g of the resulting FFS was cast onto a glass plate (350 mm × 200 mm) and dried at 45 °C in an oven for 12 h. The dried films were peeled off and conditioned at 23 ± 2 °C and 50 ± 5% relative humidity (RH) in a chamber (HD-E702-100, Dongguan Haida Equipment Co., Ltd., Dongguan, China) until further evaluation.

### 2.5. Characterization of the Films

#### 2.5.1. Fourier-Transform Infrared (FTIR) Spectroscopy

The FTIR spectra of the RCP, PVA, and RCP/PVA films were recorded from 650 to 4000 cm^−1^ with 32 scans and a nominal resolution of 4 cm^−1^ using a Nicolet IS10 spectrometer (Thermo Fisher Scientific, Waltham, MA, USA) equipped with an attenuated total reflectance (ATR) accessory.

#### 2.5.2. Scanning Electron Microscope (SEM)

The microstructures of the RCP, PVA, and RCP/PVA films were characterized by a Quanta 250 SEM (FEI Co., Ltd., Hillsboro, OR, USA) at an accelerating voltage of 20 kV. To observe the cross-sectional surface, the films were cryofractured in liquid nitrogen in advance. The samples were mounted onto the specimen holder and coated with gold before observation.

#### 2.5.3. Physical Properties

The tensile properties of the films in terms of their tensile strength at break (TS) and percentage of elongation at break (EAB) were measured by an auto tensile tester (HD-A821-1, Haida International Equipment Co., Ltd., Dongguan, China) with an initial gap of 40 mm and a speed of 1 mm/s according to the ASTM standard method, D882-2018 [38]. At least five rectangular strips (15 mm × 80 mm) were cut from the films and tested. 

The water solubility (WS) of the films was assayed as follows [39]: Square samples with a side length of 30 cm were dried at 105 °C for 24 h and weighed (*W*_1_), then immersed into 50 mL of distilled water and shaken for 24 h at 25 °C. Finally, the insoluble matter was dried at 105 °C for 24 h and weighed again (*W*_2_). The WS (%) was calculated based on the following equation:(3)$$\mathrm{WS}\left(\%\right)=\frac{W_{1}-W_{2}}{W_{1}}\times 100$$

The water vapor permeability (WVP) of the films was determined with a WVP tester (W3/031, Lab think Instruments Co., Ltd., Jinan, China) according to the standard testing method, ASTM E96-16 [40]. The detailed procedures and equations for calculating the measurements have been described in our previous work [41].

#### 2.5.4. Color Response Properties

The color response of the RCP and RCP/PVA films was evaluated by the color change at different pH conditions. Rectangular samples (40 mm × 30 mm) cut from the films were immersed into 10 mL of different pH buffer solutions (pH from 2 to 12), which were formed using a HCl aqueous solution for pH levels from 2 to 6, distilled water for a pH level of 7, and a NaOH aqueous solution for pH levels from 8 to 12 [13]. After immersion for 10 min, images of the samples were captured at the same time. The color of the RCP/PVA film corresponding to a particular pH value was measured at 5 random points using an automatic colorimeter (CM2300D, Konica Minolta Investment Co., Ltd., Tokyo, Japan). The color parameters, including *L** (lightness), *a** (redness–greenness) and *b** (yellowness–blueness), were obtained to calculate the total color difference (Δ*E*) using Equation (4):(4)$$\Delta E=\sqrt{{\left(L^{*}-L\right)}^{2}+{\left(a^{*}-a\right)}^{2}+{\left(b^{*}-b\right)}^{2}}$$
where *L*, *a*, and *b* are the original color parameter values of the as-prepared film, and *L**, *a**, and *b** are the color parameters of the film after discoloration (at different pH levels).

### 2.6. Application in Monitoring Freshness of Fish

To monitor the freshness of fish, the RCP/PVA film was cut into squares (30 mm × 30 mm) and each piece was stuck on the headspace of a sterilized Petri dish containing a 30 g fillet of fresh crucian carp, which was then stored at 25 °C for 48 h. Chemical and microbiological analyses, including the total volatile basic nitrogen (TVB-N), total viable count (TVC), and pH of the fillet during storage, were evaluated according to the methods described by Huang et al. [7] and Moradi et al. [8]. The color transformation of the film was captured simultaneously, and the color parameters were measured to calculate the ΔE values of the films according to the method described above in Section 2.5.4.

### 2.7. Statistical Analysis

The statistical analyses of the experimental data were performed using an analysis of variance (ANOVA) in SPSS software (version 13.0, SPSS Inc., Chicago, IL, USA). The data were given as mean values ± standard deviation. The significant differences among the data were determined by Duncan’s multiple range test (*p* < 0.05). 

## 3. Results and Discussion

### 3.1. Characterization of FFSs

#### 3.1.1. PSD, pH, and TAC of RCP Suspension

It should be emphasized that the main novelty of this study is related to the direct use of red cabbage as a film-forming substance and an anthocyanin source for the development of intelligent pH indicator films, since it combines the film-forming properties of naturally present biopolymers (such as cellulosic and pectic substances) with colorants (such as anthocyanins) responsible for the related purple/red color of red cabbage [29,33,42,43,44]. The particle size distribution of the homogenized RCP suspension is depicted in Figure 1. The homogeneous and purple suspension, with an average particle size (D_av_) of 12.86 ± 0.03 μm, was obtained due to the effect of high-pressure homogenization on disrupting the vegetable pulp particles. The D_av_ of the RCP suspension for film-forming fell in the range of various suspensions that have been used for preparing other biopolymer films derived from fruits and vegetables, such as cutin extracted from tomato peels (7.1 μm) [45], pomelo peel flour (19.87 μm) [39], and banana flour (31.7 μm) [46]. Additionally, the RCP suspension showed a final pH of 5.58 ± 0.01 and a TAC of 292.17 ± 2.65 mg/L. A comparable result has been reported by Chandrasekhar et al. [11] when extracting anthocyanins from red cabbage in water (pH = 5.1 and TAC = 301 mg/L). It can be concluded from the results given above that the RCP suspension, with an appropriate particle size and anthocyanin content, has the potential to be used to develop intelligent films.

#### 3.1.2. Rheological Properties of FFSs

Figure 2A shows the variation in the apparent viscosity of the FFSs as a function of shear rate. The pure PVA solution showed Newtonian flow behavior, whereas both the RCP and RCP/PVA suspensions presented a typical pseudo-plastic behavior of shear thinning (the viscosity decreased with increasing shear rate), especially in the low shear rate range (<20 s^−1^). Similar steady-flow behaviors have been observed in previous studies on PVA solutions [47,48] and other homogenized suspensions from fruits or vegetables, such as tomato juice [49], mango juice [50], and pomelo peel flour [39]. In addition, it is worth noting that as compared to the RCP suspension, the shear thinning effect of the RCP/PVA suspension was more profound after the addition of PVA, and its viscosity was higher than that of the RCP or PVA, especially when the shear rate was lower than 60 s^−1^, indicating an entangled network in the RCP/PVA suspension, induced by intermolecular interactions such as hydrogen bonding between RCP and PVA [19,28,47,49]. A similar synergistic effect of PVA on reinforcing a biopolymer network has also been reported by Ma et al. [48] for a tara gum solution blended with PVA. 

An evaluation of the dynamic rheological properties is necessary because it could reflect the structural characteristics and interactions between components of the FFS [51]. The dynamic rheological curves of the FFSs from dynamical frequency sweeps are plotted in Figure 2B–D. All the FFSs showed an increasing trend in the values of G′ (representing elastic behavior) and G″ (representing viscous behavior) with an increase in the angular frequency. For the pure PVA solution, as shown in Figure 2B, G″ was always apparently higher than G′ over the entire range of the considered angular frequencies, indicating a largely viscous, fluid-like behavior [47,52]. However, Figure 2C,D show that the RCP and RCP/PVA suspensions were weak gel systems, where the G″ values were found to be higher than the G′ values at low frequencies and the inverted order was observed at high frequencies [19]. Furthermore, the crossover point (or gel point) of the curves of G′ and G″ (at which G′ = G″) noticeably shifted to a higher angular frequency after incorporating the PVA. A similar phenomenon has also been observed in previous studies [19,28], which was mainly attributed to the formation of new intermolecular interactions and the reconstruction of an entangled network among the components of the FFS, agreeing well with the steady-flow rheological results.

#### 3.1.3. Color Variations and UV-Vis Spectra of RCP and RCP/PVA Suspensions

The color variations of the RCP and RCP/PVA suspensions were evaluated to validate their potential use as a pH-sensing dye. As displayed in Figure 3A,B, the colors of both suspensions changed from red to yellow as the pH increased from 2 to 12. Specifically, the suspensions displayed a bright red color at a pH of 2–3, changed gradually from pink at a pH lower than 5 to purple at a pH of 6–7, and turned blue, green and yellow at pH values of 8, 9–10, and 11–12, respectively. Corresponding with the above color variations, the maximum adsorption of the UV-Vis spectra for the RCP and RCP/PVA suspensions at different pH values also changed, as evidently displayed in Figure 3C,D. This phenomenon is called a bathochromic shift [53] and is commonly ascribed to the structural transformations of anthocyanins present in the FFS according to the literature [19,54,55]. For example, the RCP suspension exhibited characteristic flavylium cation absorbance at 525 nm and a pH of 2, whereas this maximum adsorption peak decreased in intensity and shifted towards longer wavelengths (ca. 530–540 nm) as the pH value increased to 6, demonstrating a notable transformation of the flavylium cations to the quinonoid bases. As the pH increased to over 7, due to the formation of a carbinol pseudobase, the maximum absorption peak shifted to a value of approximately 602 nm, and the absorbance gradually increased from a pH of 7 to a pH of 9. Furthermore, the absorbance at 602 nm decreased with an increase in the pH in the range of 10–12. Similar results for color variations and UV measurements have been previously reported for anthocyanin extract solutions from red cabbage [19,56] and purple sweet potato [56,57]. Therefore, the homogenized RCP suspension showed a sufficient pH sensitivity through color variations originating from the anthocyanins present in the sample, suggesting its potential use as a pH indicator analogous to extracted anthocyanin solutions. Additionally, as shown in Figure 3D, there was no significant change in the UV-Vis spectra of the RCP/PVA suspension when compared to the RCP suspension (Figure 3C), except for some slight reductions in the absorbance, signifying that the pH sensitivity of the FFS was not obviously affected by the incorporation of PVA.

### 3.2. Characterization of the Films

#### 3.2.1. FTIR Analysis

As shown in Figure 4, the PVA film exhibited characteristic peaks at around 3260 cm^−1^, 2924 cm^−1^, and 1084 cm^−1^, corresponding to O-H stretching, -CH stretching from methyl groups, and C-O stretching vibrations, respectively [20,24,58]. As for the RCP film, the similar bands at around 3282 cm^−1^ and 2923 cm^−1^ could be assigned to the related characteristic groups (O-H and C-H) of biopolymers, such as the cellulose and pectin present in red cabbage [39,41,59]. Moreover, there was a strong absorption band with a maximum at 1034 cm^−1^ corresponding to the anhydroglucose ring’s O-C stretching vibration, as well as bands at 1624 cm^−1^ and 1526 cm^−1^ corresponding to stretching vibrations of the C=C aromatic rings due to the presence of phenolic compounds [18,20]. In addition, another signal at 1729 cm^−1^ was also detected, which could be attributed to the presence of C=O groups from carboxylic acids [31,32]. 

In the spectrum of the RCP/PVA film, there were main characteristic absorption peaks for both the RCP and PVA as expected, but no new additional peaks appeared, suggesting that no chemical interactions occurred between them. Nevertheless, it was worthy to note that when comparing the FTIR spectrum of the RCP/PVA film with that of the RCP film, some changes in the band positions and peak intensities were observed. The band at around 3282 cm^−1^ (related to the O-H stretching vibration) became broader after the addition of PVA, suggesting the formation of hydrogen bonds between RCP and PVA [17,58]. The band at 1624 cm^−1^ (corresponding to the C=C aromatic ring stretching, which is a characteristic of anthocyanins) decreased in its intensity and shifted to a lower wavenumber, probably because of the electrostatic interactions between PVA and the anthocyanins in RCP, which was in accordance with previous findings for intelligent films based on biopolymers with incorporated red cabbage anthocyanins [13,17,18,19]. It was consequently expected that these physical interactions (in terms of hydrogen bonding and/or electrostatic interactions) between the PVA and RCP, as revealed by the FTIR analysis, would contribute to improving the structure and properties of the composite film without affecting its sensitivity to pH changes [18].

#### 3.2.2. Morphology

Figure 5 shows the SEM micrographs of the surfaces and cross-sections of the films. Consistent with a previous report [23], the PVA film had a typical smooth and homogeneous microstructure, as seen in Figure 5a,d. In contrast, the RCP film exhibited a rough surface with some insoluble aggregates of fibrous particles (Figure 5b) as well as loose cross-sectional structure with some pores (Figure 5e), probably due to the poor film-forming capability of RCP [59]. Similar observations for the micromorphology of fruit- and vegetable-based films have also been made by Andrade et al. [59] for fruit and vegetable residue flour, Tulamandi et al. [60] for papaya puree, and Wu et al. [39] for pomelo peel flour. For the RCP/PVA film, it was observed from Figure 5c,f that, in comparison to the RCP film, the surface was relatively smoother, and its cross-section seemed to be more continuous and compact to some degree, verifying the strong interaction and good compatibility of RCP with PVA [34]. Such a better homogeneity of a film is also a good indicator of its structural integrity, which is consequently beneficial for improving the mechanical and barrier properties of composite films [34,39].

#### 3.2.3. Physical Properties

Mechanical behaviors are closely related to a film’s structure and construction [60]. The TS (mechanical resistance) and EAB (flexibility) of the films obtained by the tensile test are listed in Table 1. The PVA film showed the best mechanical properties among all the films, with a TS value of 36.3 MPa and an EAB value of 265.1%, which were in accordance with those (37.5 MPa and 263.5%) reported in the literature [20]. According to Otoni et al. [29], the mechanical properties of edible films based on fruits and vegetables strongly depend on their composition, which are reported to have TS and EAB values ranging from 0.03 MPa to 30 MPa and 1.8% to 217%, respectively. In this regard, although the mechanical properties of the RCP film developed in the present study were inferior to those of the PVA film, they were comparable to those reported for edible films based on fruits and vegetables [29,60,61,62,63]. Furthermore, as shown in Table 1, the TS and EAB of the RCP/PVA film were significantly increased by 90.8% and 41.7% (*p* < 0.05), respectively, when compared with the RCP film. Such simultaneous increments in the TS and EAB of composite films could be ascribed to the stronger intermolecular interactions through hydrogen bonding between the matrix (RCP) and the binding agent (PVA) as deduced by rheology and the FTIR analysis (Figure 2 and Figure 4), as well as the more homogeneous and compact network structure with less noticeable pores as confirmed by SEM observation (Figure 5). Similar results were found for other biopolymer-based films incorporating PVA, including cellulose films [64], pea starch films [65], and tara gum films [48].

The same trend was obtained for the water resistance of the films, which was analyzed in terms of the WS and WVP. The biopolymers either naturally present in fruits and vegetables or added as binding agents usually have a high polarity, hydrophilicity, and water-holding capacity [29]. These characteristics endow such films with poor water resistance, which may be undesirable for some potential applications. For example, the WVP values of edible films based on fruits and vegetables are much higher than those of synthetic polymer films, ranging from 0.10 to 13.57 g·mm/m^2^·h·kPa [29,66]. Likewise, the pure RCP film had relatively poorer water-resistant properties than the pure PVA film, as reflected by the WS and WVP values listed in Table 1, which was concordant with the hydrophilic nature of the biopolymers (cellulose derivatives, pectin, and other components) in the RCP. Nevertheless, the RCP/PVA film exhibited a significantly lower WS and WVP (*p* < 0.05) than the RCP film. This could also be explained by the results of the FTIR and SEM studies showing that the formation of hydrogen bonds between the hydrophilic groups of biopolymers in the RCP and the hydroxyl groups of the PVA produced a more homogeneous and dense film structure, leading to decreased water affinity and permeability and thus a better water resistance of the composite film [17,41,65]. The abovementioned results suggest that the incorporation of PVA into the RCP matrix could improve the water resistance and mechanical properties of the film, which would be conducive to its potential food-packaging application.

#### 3.2.4. Colorimetric Response

Figure 6 shows the visible color changes of both the RCP and RCP/PVA films when submerged in different buffer solutions, ranging from red-pink to green and yellow with an increasing pH from 2 to 12. The color response of the films to pH variation was basically consistent with that of the FFS (Figure 3), owing to the structural transformation of anthocyanins [17,19,20,25]. It is noteworthy that the RCP film was found to dissociate into pieces after a gentle shake when immersed in a buffer solution for a few minutes (Figure 6b), whereas the RCP/PVA film could be easily picked out and remained intact (Figure 6d), suggesting an improved water resistance [25]. Based on the above results, the addition of the PVA endowed the film with more excellent physical properties while maintaining a comparable color response, so the RCP/PVA film was selected for further exploration of an intelligent pH indicator film in the following study.

The color parameters, including *L*, *a*, *b*, and Δ*E*, of the RCP/PVA film after immersion in different pH buffer solutions were detected and are summarized in Table 2. The as-prepared RCP/PVA film had an original purple color due to the anthocyanins contained in red cabbage, which was used as a reference to calculate the total color difference (ΔE). At a pH ≤ 6, parameter *a** presented with positive values, which increased with decreasing pH, indicating a brighter red color at a lower pH [14]. At a higher pH (ranging from 7 to 12), the value of parameter *a** dramatically decreased below zero with greater alkalinity and reached a minimum of −36.23 at a pH of 10, indicating a great enhancement in greenness with increasing pH. Parameter *b** had positive values, indicating a greater tendency to yellow, especially at a higher pH (11–12). Additionally, a Δ*E* with a value bigger than 5 indicated that the color change of the RCP/PVA film could be readily detected by the human eye across a wide pH range [14,19,23]. These results were consistent with the photographs of the film after the color changes (Table 2). A similar pH sensitivity has been also reported in other biopolymer films incorporated with anthocyanins extracted from red cabbage [13,14,15,16,17,18,19,20]. However, it should be highlighted that the present color change results of the obtained film using homogenized red cabbage puree blended with PVA in this work were analogous to those found in the aforementioned studies for films containing anthocyanins extracted from red cabbage, also demonstrating its potential application as a pH indicator for intelligent food packaging, regardless of the pretreatment of anthocyanin extraction. Luchese et al. [31,32] directly incorporated blueberry residue powder into starch to develop pH indicator films, and observed visual color changes after immersion of the films in different buffer solutions. The authors attributed this effect to the leaching of water-soluble color compounds such as anthocyanins from the blueberry residue powder present in the film into the water [32]. In another study [30], pH-sensitive films were prepared by incorporating *Prunus maackii* juice as a raw material into a κ-carrageenan/hydroxypropyl methylcellulose composite matrix. Consequently, red cabbage puree, which is also rich in anthocyanins, can be an alternative pH indicator for the development of intelligent films to perceive food deterioration, as will be demonstrated in the latter part of this work.

#### 3.2.5. Application in Monitoring Fish Freshness

In consideration of the pH-sensing characteristics, the RCP/PVA film was used to monitor fish spoilage as determined by freshness indicators, including TVC, TVB−N, and pH. Microbial activity and other biochemical reactions are the main reasons for fish spoilage [8,57]. As shown in Figure 7A, an increase in the TVC of fish was accompanied by an increase in the pH and TVB-N during storage at 25 °C. According to previous works [7,57], the rejection limits of the TVC and TVB−N levels for fresh fish products are 7 lg CFU/g and 20 mg/100 g, respectively. The initial values for the TVC and TVB−N of fish were 4.95 lg CFU/g and 6.30 mg/100 mg, respectively, indicating the high-quality freshness of the samples [8]. In our study, the TVC increased to 7.62 lg CFU/g after 24 h of storage, exceeding the mentioned threshold value. For TVB-N, its value was 13.49 mg/l00 g at 24 h and increased to 25.99 mg/l00 g at 36 h, indicating that the fish was not fresh. Notably, there was a time difference between the elapsed times required to reach the TVB−N and TVC thresholds. This lag phenomenon can be attributed the fact that the accumulation of a volatile base results from an inherent increase in microbial activity and population size, which has been previously observed by other authors [7,67]. Besides, the pH value of fish gradually increased from an initial 6.19 to 7.91 after storage for 48 h, which could be attributed to microbial metabolites as well as endogenous enzymes, elevating microbial proliferation and leading to an inferior quality [8,9]. Similar results have been reported in earlier studies for fish samples during storage at 25 °C [7,57,68]. Based on the thresholds and measured values of the TVC and TVB−N, it can be consequently determined that the initial stage of fish spoilage at 25 °C occurred after about 18–30 h, similar to the results obtained by Huang et al. [7].

The color changes in the RCP/PVA film during fish storage at 25 °C are displayed in Figure 7B. As shown, the RCP/PVA film transformed from purple to purplish blue at 24 h, and eventually became yellow after 36 h or more. In accordance with the above color changes, the *a** value of the RCP/PVA film decreased while the *b** and Δ*E* values of the film increased with increasing storage time until 36 h. In particular, based on previous studies [7,14], Δ*E* values greater than 5 can be detected by the human eye, and values over 12 signify an absolute difference in color that is very noticeable even by untrained panelists. The Δ*E* value of the RCP/PVA film was 8.70 after a storage time of 12 h, and reached 26.04 after 24 h, implying that the color change would be easily detectable by a consumer. The visible color transition point (Δ*E* > 12) at around 24 h was consistent with the above fish spoilage stage established in Figure 7A. Therefore, these results suggest that the developed film based on anthocyanin-rich red cabbage puree and PVA could monitor real-time fish freshness via a visible color change, which could be utilized as a colorimetric pH indicator film in intelligent packaging.

## 4. Conclusions

A novel pH-sensing biopolymer film was fabricated by blending red cabbage puree with PVA via a solution-casting method. The homogenized RCP suspension (with a D_av_ of 12.86 ± 0.03 μm and a TAC of 292.17 ± 2.65 mg/L) was directly used as a film-forming substance and a source of anthocyanins without prior extraction. The rheological and FTIR results revealed that there were physical interactions (in terms of hydrogen bonding and/or electrostatic interactions) between the RCP and PVA, resulting in a more homogeneous and compact film microstructure, as confirmed by SEM. Consequently, the addition of PVA improved the mechanical properties and water resistance of the RCP film, but had no significant effect on the pH sensitivity. The color of the film changed significantly from red to yellow in solutions with pH values of 2 to 12. The application trial suggested that the developed RCP/PVA film could monitor fish freshness via a visible color change corresponding to the fish spoilage point. The results highlight the potential use of anthocyanin-rich purees of fruits and vegetables such as RCP, a feasible alternative to anthocyanins extracted from them, in developing eco-friendly pH-sensitive indicator films with a low production cost for intelligent food packaging. Although such films were produced successfully in this study, further studies are required to optimize the formulations and performance of the films for more foodstuffs in the future.

## Figures and Tables

**Figure 1 foods-11-03371-f001:**
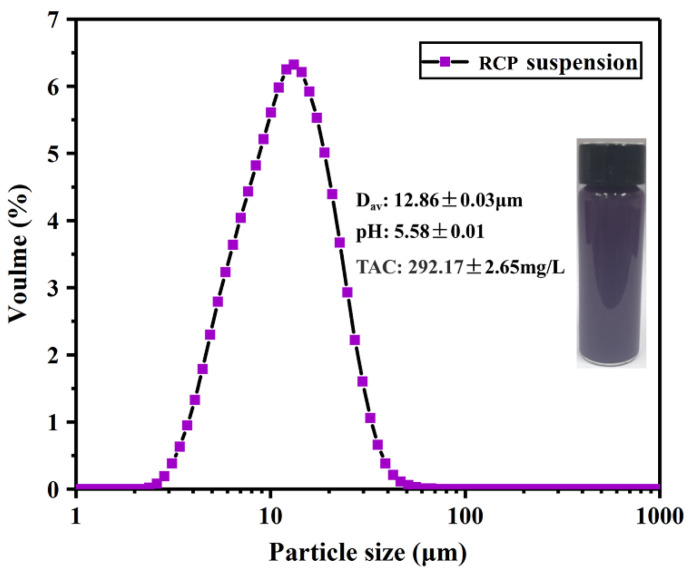
Visual appearance, particle size distribution (PSD), total anthocyanin concentration (TAC), and pH of RCP suspension.

**Figure 2 foods-11-03371-f002:**
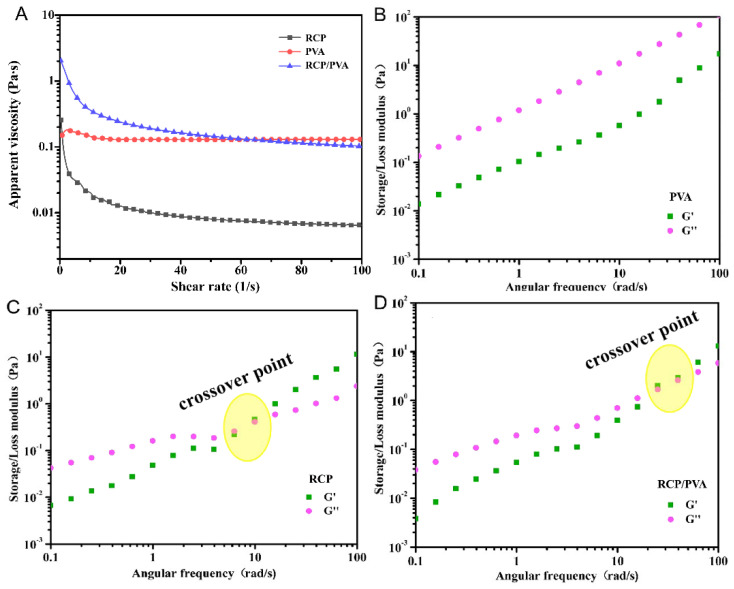
Steady (**A**) and dynamic (**B**–**D**) rheological properties of the FFS.

**Figure 3 foods-11-03371-f003:**
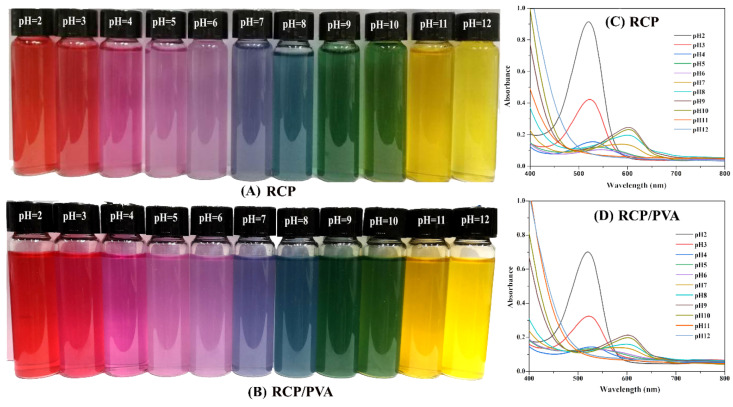
Color variations (**A**,**B**) and UV−Vis spectra (**C**,**D**) of the RCP and RCP/PVA suspensions in a pH range of 2–12.

**Figure 4 foods-11-03371-f004:**
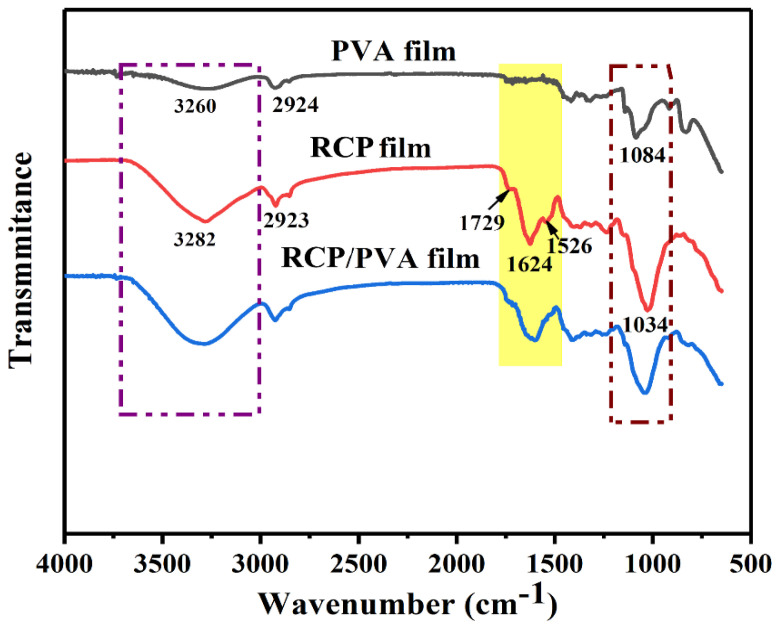
FT−IR spectra of PVA, RCP, and RCP/PVA films.

**Figure 5 foods-11-03371-f005:**
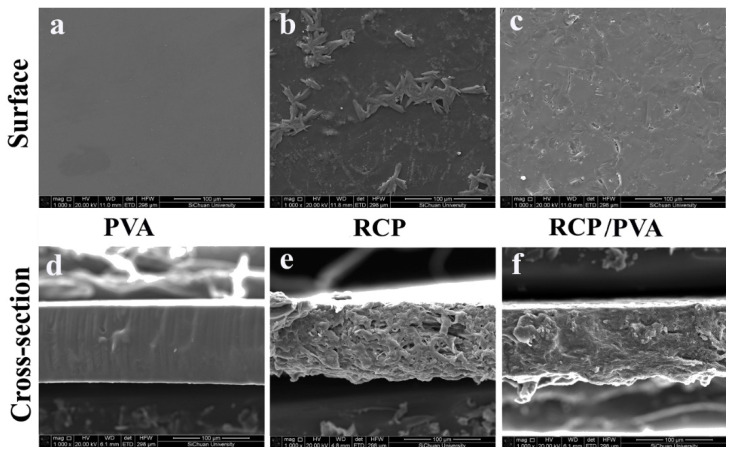
Micrographs of the surfaces (**a**–**c**) and cross sections (**d**–**f**) of PVA, RCP, and RCP/PVA films.

**Figure 6 foods-11-03371-f006:**
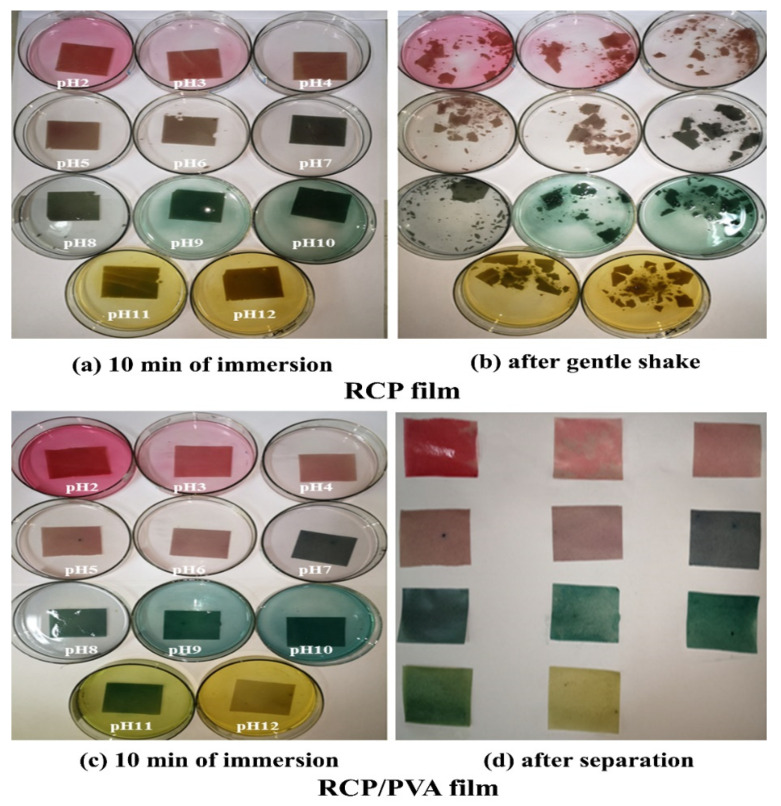
Colorimetric response of RCP and RCP/PVA films immersed in different pH buffer solutions (pH of 2–12).

**Figure 7 foods-11-03371-f007:**
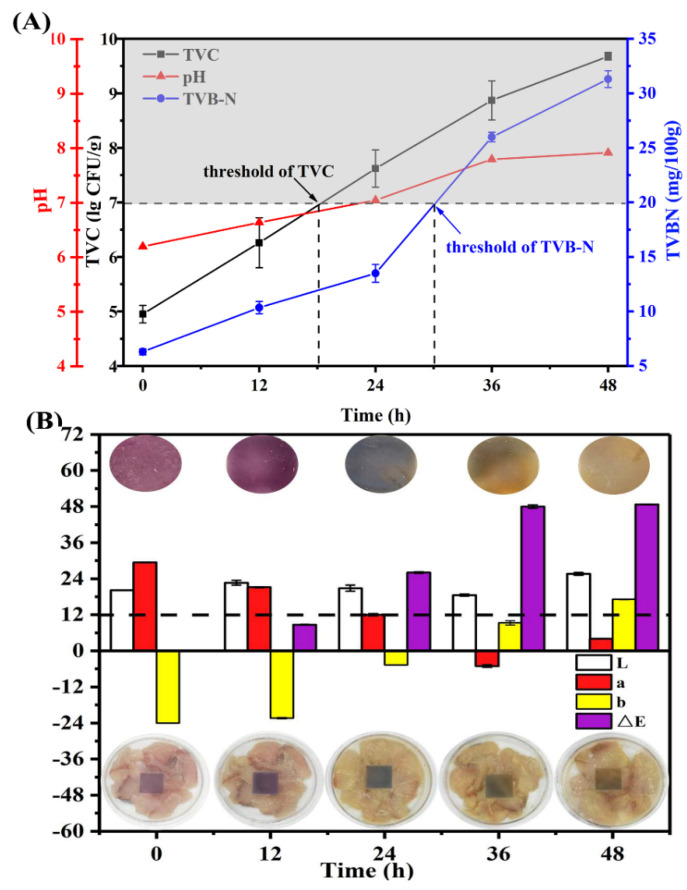
Changes in TVB−N, TVC, and pH values of fish (**A**) and the color response and parameters (**B**) of RCP/PVA film during storage at room temperature (25 °C).

**Table 1 foods-11-03371-t001:** Mechanical properties and water resistance of the PVA, RCP, and RCP/PVA films.

Film	TS (MPa)	EAB (%)	WS (%)	WVP (g·mm/m^2^·h·kPa)
PVA	36.3 ± 1.9 a	265.1 ± 23.1 a	7.5 ± 0.1 c	0.97 ± 0.06 c
RCP	6.5 ± 0.4 c	14.4 ± 2.0 b	43.5 ± 3.0 a	1.60 ± 0.09 a
RCP/PVA	12.4 ± 0.3 b	20.4 ± 1.4 b	33.0 ± 0.2 b	1.20 ± 0.15 b

Different letters within the same column indicate significant differences (*p* < 0.05).

**Table 2 foods-11-03371-t002:** Color parameters of the RCP/PVA film after immersion in different pH buffer solutions.

pH Value	Color	*L**	*a**	*b**	Δ*E*
origin	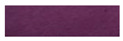	55.46 ± 1.08 f	31.06 ± 0.78 b	5.50 ± 0.29 f	—
pH 2	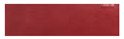	55.77 ± 0.36 f	54.04 ± 0.46 a	18.83 ± 0.16 c	26.6 ± 0.81 b
pH 3	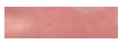	71.22 ± 0.58 b	23.67 ± 2.80 c	14.26 ± 0.63 d	19.6 ± 2.41 d
pH 4	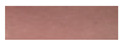	70.18 ± 0.21 b	20.33 ± 0.13 d	14.41 ± 0.38 d	20.28 ± 1.24 d
pH 5	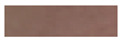	65.72 ± 0.93 c	13.76 ± 0.34 e	14.04 ± 0.19 d	21.87 ± 1.90 d
pH 6	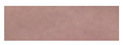	61.39 ± 0.05 d	9.84 ± 0.07 f	10.50 ± 0.07 e	22.60 ± 1.00 cd
pH 7	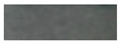	51.07 ± 0.45 g	−6.43 ± 0.08 h	3.94 ± 0.20 g	25.09 ± 0.47 bc
pH 8	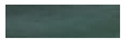	46.74 ± 0.18 h	−12.81 ± 0.37 i	2.64 ± 0.12 g	20.46 ± 0.35 d
pH 9	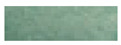	51.5 ± 0.74 g	−36.18 ± 0.60 k	12.90 ± 0.31 d	9.97 ± 1.32 e
pH 10	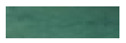	49.75 ± 0.61 i	−36.23 ± 0.73 k	13.22 ± 0.10 d	10.96 ± 1.32 e
pH 11	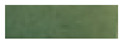	57.6 ± 0.09 e	−18.73 ± 0.59 j	26.49 ± 2.77 b	24.88 ± 2.07 bc
pH 12	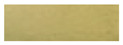	73.24 ± 1.42 a	−3.81 ± 0.06 g	39.41 ± 1.13 a	47.02 ± 1.17 a

Different letters within the same column indicate significant differences (*p* < 0.05).

## Data Availability

The data presented in this study are available from the corresponding author upon request.

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
