# Peer review of "A Facile Strategy for Development of pH-Sensing Indicator Films Based on Red Cabbage Puree and Polyvinyl Alcohol for Monitoring Fish Freshness"

_foods, 2022, doi:10.3390/foods11213371_

Round 1

Reviewer 1 Report

The paper 'A facile strategy for development of pH sensing indicator films based on red cabbage puree and polyvinyl alcohol for monitoring fish freshness' describes a novel pH sensing biopolymer film based on red cabbage puree incorporating polyvinyl alcohol, which was then used to monitor the fish freshness. Authors described the preparation and properties of the obtained films.

The subject matter is up-to-date, innovative, in line with the current trends in science, with great potential application. The article contains extensive and interesting experimental material.

1. What amounts of HCl and NaOH were used to achieve the stated pH values (section 2.3.4). Would it not be reasonable to use model acids and bases found in food products?

2. Please indicate which buffer solutions were used to achieve the pH values listed (section 2.5.4).

3. Is the color of film-forming suspensions (FFS) and films stable over time? Has the color been measured at different time intervals?

Reviewer 2 Report

The subject of this work is not so new. Nevertheless, this work provides useful information, is sufficiently complete, well documented or referenced and comprehensive, clearly written and executed based on appropriate, and reliable technologies. The discussion of results is correct and clear. From the foregoing, I consider that this manuscript could be accepted for publication, however, some observations that must be addressed are indicated below.

On line 122 it says: “RCP suspension were prepared by adding…”. I think it should say: “RCP suspension was prepared by adding…”.

On line 157 it says: “…FD is the dilution factor,…”. I think it should say: “…DF is the dilution factor,…”.

On lines 149-150 it says: “… and their absorbance were measured at 530 and 700 nm by a multi-mode microplate reader…”. First of all I think it should say: “… and their absorbance was measured at 530 and 700 nm by a multi-mode microplate reader…”. On the other hand, this statement is confusing or unclear, since formula 1 shown on line 154 indicates:

A = (A510 – A700) pH = 1 – (A510 – A700) pH= 4.5

On line 247 it says: “…was obtained due to the effect of HPH in disrupting the vegetable pulp…”. I think this statement is unclear, since the meaning of the term HPH was not previously indicated or defined.

The titles of figures and tables are difficult to understand, since the meaning of the acronyms and abbreviations used are not indicated on the figure or table itself. I consider that any figure or table should be understood by itself, that is, without the need to read the text.
